# A Novel Rabbit Model of Retained Hemothorax with Pleural Organization

**DOI:** 10.3390/ijms25010470

**Published:** 2023-12-29

**Authors:** Christian J. De Vera, Rebekah L. Emerine, René A. Girard, Krishna Sarva, Jincy Jacob, Ali O. Azghani, Jon M. Florence, Alan Cook, Scott Norwood, Karan P. Singh, Andrey A. Komissarov, Galina Florova, Steven Idell

**Affiliations:** 1Department of Cellular and Molecular Biology, School of Medicine, The University of Texas Health Science Center at Tyler, 11937 US HWY 271, Tyler, TX 75708, USA; christianjordan.devera@uttyler.edu (C.J.D.V.); remerine@patriots.uttyler.edu (R.L.E.); rene.a.girard@uth.tmc.edu (R.A.G.); krishna.sarva@uttyler.edu (K.S.); jj12581@uttyler.edu (J.J.); jon.florence@uttyler.edu (J.M.F.); andrey.komissarov@uttyler.edu (A.A.K.); galina.florova@uttyler.edu (G.F.); 2Department of Biology, The University of Texas at Tyler, 3900 University Blvd, Tyler, TX 75799, USA; aazghani@uttyler.edu; 3Department of Surgery, School of Medicine, The University of Texas Health Science Center at Tyler, 11937 US HWY 271, Tyler, TX 75708, USA; alancook@uttyler.edu (A.C.); scottie.norwood@uttyler.edu (S.N.); 4Department of Epidemiology and Biostatistics, School of Medicine, The University of Texas Health Science Center at Tyler, 11937 US HWY 271, Tyler, TX 75708, USA; karan.singh@uttyler.edu

**Keywords:** retained hemothorax, pleural organization, fibrinolysis, thoracostomy tube, coagulation, pleural fluid, rabbits

## Abstract

Retained hemothorax (RH) is a commonly encountered and potentially severe complication of intrapleural bleeding that can organize with lung restriction. Early surgical intervention and intrapleural fibrinolytic therapy have been advocated. However, the lack of a reliable, cost-effective model amenable to interventional testing has hampered our understanding of the role of pharmacological interventions in RH management. Here, we report the development of a new RH model in rabbits. RH was induced by sequential administration of up to three doses of recalcified citrated homologous rabbit donor blood plus thrombin via a chest tube. RH at 4, 7, and 10 days post-induction (RH4, RH7, and RH10, respectively) was characterized by clot retention, intrapleural organization, and increased pleural rind, similar to that of clinical RH. Clinical imaging techniques such as ultrasonography and computed tomography (CT) revealed the dynamic formation and resorption of intrapleural clots over time and the resulting lung restriction. RH7 and RH10 were evaluated in young (3 mo) animals of both sexes. The RH7 recapitulated the most clinically relevant RH attributes; therefore, we used this model further to evaluate the effect of age on RH development. Sanguineous pleural fluids (PFs) in the model were generally small and variably detected among different models. The rabbit model PFs exhibited a proinflammatory response reminiscent of human hemothorax PFs. Overall, RH7 results in the consistent formation of durable intrapleural clots, pleural adhesions, pleural thickening, and lung restriction. Protracted chest tube placement over 7 d was achieved, enabling direct intrapleural access for sampling and treatment. The model, particularly RH7, is amenable to testing new intrapleural pharmacologic interventions, including iterations of currently used empirically dosed agents or new candidates designed to safely and more effectively clear RH.

## 1. Introduction

Retained hemothorax (RH), a residual collection of blood persisting after 72 h of attempted chest tube drainage [1], is a commonly encountered consequence of intrapleural hemorrhage that is associated with early organization of the retained clot [2]. Pleural bleeding arises in penetrating or blunt thoracic trauma, which occurs in 20–25% of all trauma patients and contributes to about 30% of trauma-related mortality [1,3]. It also occurs after thoracic surgery and with bleeding diatheses associated with malignancy, drugs, or collagen vascular diseases [2]. RH is commonly associated with thickening of the visceral pleural surface and adhesions with proximate pleural symphysis that extend from the organizing fibrinous collection to the parietal pleura and lung, resulting in impaired expansion or entrapment [2,4]. Depending on the size of the RH, extent of pleural organization, and limitation of lung expansion, dyspnea can ensue. More than a fourth of patients with traumatic RH can develop empyema [5,6,7,8]. If tube thoracostomy, often the first choice of treatment [9], fails in RH, patients are left with video-assisted thoracoscopic surgery (VATS) and lytic therapy as potential options [10,11,12,13]. Since delayed drainage increases the risk [14] of poor outcomes for these patients [15], early intervention after recognition of RH has been advocated [4,16]. VATS has been recommended as the initial intervention for RH, where blood in the pleural space is clotted and loculated [9,17]. The risks of this approach are due to its invasiveness, requirement for general anesthesia, and the possibility of complications of the procedure itself. These considerations have led to the utilization of intrapleural fibrinolytic therapy (IPFT) as an alternative strategy [2,4,12,17,18,19]. There is literature supporting the use of IPFT to clear RH, which may be of value in poor surgical candidates or those that have underlying advanced malignancy or other co-morbidities [4,17,18]. While the use of IPFT is inherently appealing as a pharmacologic alternative to surgery, the approach is disadvantaged by the paucity of literature defining the optimal fibrinolytic agents to be used for RH, their dosing, and their schedules of administration, as is the case in empyema [19,20]. While understudied, one report suggested that the addition of DNase to tPA did not improve the outcomes of RH [21]. Whether the same forms of IPFT should be used for cases of RH as for empyema remains unclear.

A novel swine model [22] recapitulates the early development of pneumothorax, hemothorax, and hemopneumothorax, but does not simulate RH for preclinical studies as it is limited to 2 h of interventional opportunity. Alternatively, fibrinolytic proteins in New Zealand White rabbits bear a close resemblance to those in humans and have been utilized extensively in IPFT preclinical studies with human plasminogen activators [20]. Previous rabbit models of chemical [23] and bacterial-induced pleural injury were reported [24,25,26,27,28,29], demonstrating their suitability for IPFT testing. Here, we report the successful development and validation of a novel model of RH in rabbits, thereby addressing a crucial translational gap in the field.

## 2. Results

### 2.1. Experimental Design

As depicted in Figure 1, hemothorax was induced on day 0 by intrapleural injection of 30 mL of citrated blood collected from donors supplemented with human thrombin (0.3 U/mL) and CaCl2 (10 mM) delivered by chest tube. Clot formation and retention were monitored via ultrasonography. Chest tube patency was maintained by daily saline injection for up to 8 days. Three intervals (4, 7, and 10 days; RH4, RH7, and RH10, respectively) were selected for optimization of the RH model. One (RH4) or two (RH7 and RH10) additional injections of blood (60 and 30 mL, respectively) were performed at 24 h (60 mL) and at 72 h (30 mL) after initiation of the models. A total of 68 rabbits were committed to RH induction, and 75 more animals were used as donors of homologous blood. Exsanguination was performed by direct intracardiac puncture or thoracotomy on donor rabbits. The first RH group was established for four days (RH4) in young female rabbits (*n* = 18, 3 mo; ~3 kg) with two blood injections totaling 90 mL. Next, to determine clot retention and organization over more extended periods, the model was allowed to progress to 7 (RH7) and 10 (RH10) days (Figure 1). Young male (*n* = 8) and female (*n* = 22) rabbits and aged female rabbits (*n* = 5, 16 mo; ~5 kg) were committed to RH7, and two groups of young females (*n* = 8) and male (*n* = 7) rabbits were committed to RH10 with three blood injections, totaling 120 mL (Figure 1). The overall survival rate of rabbits with RH was 89.7% (*n* = 61), with inadvertent chest tube misplacement accounting for the few deaths (*n* = 7) observed and excluded from the groups (Table 1). Chest tube placement and lung restriction were evaluated at baseline and before euthanasia via computed tomography (CT). Gross changes within the pleural space, RH, and pleural fluids (PFs) were evaluated postmortem. PFs and lung tissue were collected for biochemical and histological analysis.

### 2.2. Progression of RH in Rabbits

Initial experiments were performed using young female rabbits (3 mo). The primary outcomes (clot size and retention, pleural adhesions) determined the sample size (*n* = 6) per group while pleural thickening was evaluated as a secondary outcome. CT documentation of the progression of thoracic injury and lung restriction is illustrated in Figure 2a. CT imaging of the chest of the RH4 or RH7 models revealed opacification of the ipsilateral hemothoraces after induction (Figure 2a(B,C)). The intrathoracic positioning of the chest tube was confirmed by CT imaging. The chest tube was maintained for up to 8 days by daily saline flushes (1.2 mL), suture, bandage, and vest placement. Symptomatic iatrogenic pneumothoraces were resolved by aspiration through the indwelling chest tube or by needle aspiration. Resolution of pneumothorax was observed in most rabbits 7 days after induction of RH (Table 1). Induction of RH was associated with decreased total lung volumes at day 4, which partially resolved by day 7 (Figure 2a(B,C)). These changes are likely in part attributable to compensatory hyper-expansion of the contralateral lung observed in 40% of the subjects by chest CT imaging (Figure 2a(D)). Chest ultrasonography revealed RH formation and confirmed the development of adhesions at each interval (Figure 2b(A–C)). The RH7 group demonstrated the largest retained clot among the RH models at each endpoint (*p* < 0.05, Figure 2b(D)). RH was detectable at 4, 7, and 10 days after induction, with identification of retained clot by gross examination (Figure 2c(A–C), yellow arrows), confirming the ultrasonographic findings. Intrapleural organization was consistently observed with adhesion formation emanating from the clot to the mediastinum, diaphragm, or chest wall (Figure 2c(A–C)). Pleural adhesions, indicative of organizing pleural injury, were more routine in the RH7 models (*p* < 0.01, Figure 2c(D)). Pleural effusions large enough to be collected were found in 70.5% of rabbits (Table 1). Pleural thickening (Figure 2d(B1–D1), black arrows) and overexpression of collagen (Figure 2d(B2–D2)) in the pleural lining were universally observed among the RH4, RH7, and RH10 models compared to that of naïve rabbit lungs from donor rabbits or the contralateral non-injured pleural tissues of rabbits with RH (Figure 2d(A1,A2)). Pleural thickening in the RH4 (*p* < 0.05) and RH7 groups (*p* < 0.0001) was significantly more pronounced than in naïve controls. RH10 rabbits demonstrated a nonsignificant increase in pleural thickening (Figure 2d(E)).

### 2.3. Effects of Sex and Age on the Development of RH in Rabbits

Based on analysis of parameters in the RH4, RH7, and RH10 models (Figure 2), we concluded that RH7 was the most representative of clinical RH. The RH10 had smaller retained clot and reduced pleural effusion volumes. Given these considerations and to conserve animals, we assessed the effects of sex in the RH7 model. As aged male animals are commercially unavailable, we compared aged (16 mo) female rabbits versus young (3 mo) male and female rabbits. Detection of RH by chest CT opacifications (Figure 3a(A–D)) and ultrasonography (Figure 3b(A–D)) was comparable between the three groups. Ultrasonographic 2D clot area assessments of representative RH collections in vivo were comparable over time in the aged and young rabbits (Figure 3b(D)). The gross appearance of the RH collections within the thoraces of young and aged female rabbits was comparable; however, pleural adhesions were significantly lower in aged females than that in young females (Figure 3c(A–D)). Pleural thickening was uniformly observed in young and aged rabbits, and the degree of visceral pleural thickening did not significantly differ between these groups (Figure 3d(A1–C1,A2–C2,D)). The contribution of age and sex to the cellular, inflammatory, and biochemical profiles of RH at 7 d is shown in Figure 4 and Figure 5. RBCs (Figure 4A) in the PFs were significantly higher in aged females (*p* < 0.05), while WBCs (Figure 4B) were comparable among the groups. Neutrophil (Figure 4C) and lymphocyte (Figure 4D) counts were significantly higher in the PFs of young females (*p* < 0.05). Basophil (Figure 4E) were comparable among the groups. There was a significantly higher macrophage/monocyte count in young male PFs (*p* < 0.001, Figure 4F). Aged females showed significantly lower levels of active PAI-1 in PFs than young male rabbits (*p* < 0.01, Figure 5A). Total plasminogen activator inhibitor 1 (PAI-1) (Figure 5B), TNF-α (Figure 5D), of IL-6 (Figure 5E) and TGF-β (Figure 5F) levels in the PFs were comparable between all groups. Aged females had a higher level of IL-8 (*p* < 0.05, Figure 5C) than those in the young female group. Glucose (Figure 5G), lactate dehydrogenase (LDH) activity (Figure 5H), and total protein (Figure 5I) were comparable between eac Overall, RH in the RH7 age and sex groups was comparable but included some detectable differences in the PF biochemical and inflammatory profiles.

### 2.4. Comparison of Rabbit Model and Human Traumatic Hemothorax PF Markers

To determine if PFs from rabbits with RH and patients with traumatic hemothorax had common characteristics, we next compared the PF cell counts and inflammatory biomarkers within rabbit and human PFs. PFs from patients with traumatic hemothoraces, of which 7/20 had attributes of RH, were used under a protocol approved by the UT Health East Texas IRB (Table 2). Human RH PFs and those with hemothorax lacking RH attributes were mostly comparable. Therefore, the patient data were subsequently combined and depicted as hHTX during analysis with the rabbit RH PF data. Total RBC counts of RH4, RH7, and hHTX PFs were comparable, unlike the RH10 PFs, which were significantly lower (*p* < 0.05, Figure 6A). Total WBCs were comparable in the PFs of RH4, RH7, RH10, and hHTX PFs (Figure 6B). Neutrophil counts at RH10 declined significantly versus those at RH4 (*p* < 0.05) and were likewise decreased versus hHTX PFs (*p* < 0.01, Figure 6C). Lymphocyte (Figure 6D) and basophil counts (Figure 6E) were comparable to hHTX PFs. Macrophage/monocyte counts in RH4 were also similar to hHTX PFs (Figure 6F). Conversely, significantly higher macrophage/monocytes counts were found in RH7 and RH10 than in hHTX PFs (*p* < 0.001, Figure 6F).

As PAI-1 promotes pleural organization and is the major inhibitor of intrapleural plasminogen activation and fibrinolysis [30,32], we next compared the levels of PAI-1 antigen and its activity in PFs of the subjects rabbit RH and hHTX, as previously reported. We found that levels of active and total PAI-1 were elevated, approaching those previously reported in rabbits with *S. pneumoniae*-induced empyema (Figure 7A,B) [14]. Notably, total PAI-1 levels (Figure 7B) in PFs of rabbit RH were significantly higher than those in hHTX PFs; however, PAI-1 activity (Figure 7A) levels were comparable between RH7, RH10, and hHTX PFs. PAI-1 activity was significantly lower in RH4 than that in hHTX PFs (Figure 7A).

Among inflammatory biomarkers, IL-8 in all rabbit models and hHTX PFs were comparable (Figure 7C). TNF-α in the RH7 and RH10 PFs was comparable to the hHTX PFs unlike the RH4 model in which TNF-α was significantly elevated (*p* < 0.0001, Figure 7D). IL-6 levels in the RH4 (*p* < 0.05) and RH7 PFs (*p* < 0.01) were significantly reduced compared to hHTX PFs. The RH10 demonstrated a similar trend but was not statistically significant (*p* > 0.05, Figure 7E). TGF-β in the RH4 (*p* < 0.05), RH7 (*p* < 0.0001), and RH10 (*p* < 0.05) were also significantly reduced compared to levels in hHTX PFs (Figure 7F). Lastly, we found that PF glucose concentrations (RH4, *p* < 0.05; RH7, *p* < 0.0001; RH10, *p* < 0.01; Figure 7G) and PF LDH activities (RH4, *p* > 0.05; RH7, *p* < 0.0001; RH10, *p* < 0.05; Figure 7H) in the RH4, RH7, and RH10 PFs were reduced versus hHTX PFs. Total protein of RH4 (*p* < 0.0001) and RH7 (*p* < 0.0001) was significantly lower than that in hHTX PFs (Figure 7I).

## 3. Discussion

Our objective in this project was to establish an RH model in the adult rabbit. Our data demonstrate that induction of RH can reliably be achieved in rabbits with administration of three doses of homologous rabbit donor blood (Figure 1). Each rabbit costs about $450, relatively less than larger animals, as are the general costs of housing and maintenance. RH was induced in most animals, attesting to the efficiency of the protocol and its reliability. The data in Table 1 are aggregate data that support this assertion, and successful induction of RH and thoracostomy tube placement improved as we accrued more experience. These features promote animal conservation. The rabbit is large enough to tolerate tube thoracostomy, with durable patency established over 7 d. The chest tube offers the advantages of simulating the generally used scenario in patients with RH, or who are often initially treated with tube thoracostomy to attempt drainage of the HTX. Evacuation of symptomatic pneumothoraces via the chest tube was also found to be effective (Table 1).

The model was also characterized by pleural organization with adhesions to local structures thus recapitulating clinical findings in RH (Figure 2c(A–D), yellow arrows). Resorption of a single dose of intrapleural homologous rabbit blood was detected in preliminary RH4 experiments, which necessitated the continuous intrapleural administration of blood. Notably, this preclinical scenario correlates with common clinical circumstances in which large amounts or sustained deposition of blood in the pleural space leads to formation of RH [2,30]. Ultimately, it became apparent that the rabbits could tolerate a total of 120 mL of intrapleural homologous blood to induce a more pronounced RH. Regardless of the volume of homologous blood delivered to RH4 (90 mL), RH7 (120 mL), and RH10 (120 mL) models, pleural adhesions were observed in each group (Figure 2c(A–D), yellow arrows). While overall lung volume was relatively preserved, pleural thickening occurred in all models (Figure 2d(A1–D1), black arrows). We infer that contralateral lung expansion, which we commonly observed during progression of RH, mitigated against decrements of overall lung volume in the model. A large and organized RH, particularly when associated with extensive adhesions, impairs pleural drainage, and often it is used to justify IPFT or surgery to relieve symptoms and prevent fibrothoraces [2,9,17,33,34,35,36,37]. Based on pleural organization, 2D clot area, pleural adhesions count, pleural thickening, and maintenance of the chest tube, we infer that RH7 is the optimal rabbit model for the future evaluation of pharmacological candidates and dosing schedules of current IPFT options.

The rabbit RH7 model exhibited minor effects based on age and sex (Figure 3, Figure 4 and Figure 5). Because older male rabbits are not commercially available, we only used female rabbits to determine the effects of aging, which represents a limitation of the current study. We cannot exclude the possibility that greater differences might occur at 10 days or more associated with age or male sex, but the common presence of RH, visceral pleural responses, and comparable pleural organization in RH4, RH7 or RH10 groups suggest that any subsequent age or sex differences are most likely to be subtle.

PF levels of total PAI-1, IL-8, IL-6, TNF-α, TGF-β, glucose, LDH activity and total protein did not significantly differ (*p* > 0.05, Figure 5) between young and aged females and young males; Figure 5. Interestingly, aged female RH7 rabbits had lower levels of active PAI-1 in PFs than young rabbits (Figure 5A). This effect could possibly be attributable to an age-dependent varying response to RH. While the physiological ramifications of this difference in PAI-1 activity remains unclear, a trend toward decreased PF PAI-1 (Figure 5A) and pleural adhesion count was observed in aged females (Figure 3c(D)). Pleural thickness did not significantly differ (Figure 3d(D)). Thus, a decrease in intrapleural active PAI-1 may contribute to a decrease in the overall severity of pleural injury associated with RH or vice versa [30]. Notably, PAI-1 was recently reported as a biomarker of the severity of septations in human empyema [38]. These results support the concept of PAI-1 as a molecular target in a range of pleural injuries [14,30]. PAI-1-Targeted Fibrinolytic Therapy (PAI-1-TFT) resulted in an increase in the efficacy of both scuPA and sctPA in chemically induced pleural injury [30], lending further support to the concept of targeting PAI-1 to improve outcomes of organizing pleural injuries including but not limited to RH.

The rabbit PFs demonstrated elevated red blood cell counts typical of sanguineous effusions seen in clinical RH (Figure 6A). The typical mixed white cell counts generally seen in PFs of patients with RH were also identified in the PFs from the model (Figure 6B). Differences in the differential cell counts between hHTX and RH PFs may reflect the variable times at which the human samples were harvested, variable clinical co-morbidities, and possible differences in the time taken to harvest and process the clinical versus the preclinical PF samples. Because the model involves chest tube placement and because the homologous blood has the potential to elicit an inflammatory response, mediators of pleural inflammation were determined in PFs as previously described (Figure 5 and Figure 7) [14]. The presence of high levels of TGF-β in RH PFs is of particular interest given its central role in the induction of PAI-1 by mesothelial and other resident cell types and its involvement in the pathogenesis of pleural organization [14]. Levels of TGF-β in PFs from hHTX were statistically higher than those in RH4, RH7, or RH10 (*p* < 0.05, Figure 7F) and the differences between the amounts present in rabbit RH versus hHTX PFs may reflect the same influences that could have impacted differences in the cellular components of the PFs.

The levels of total PAI-1 in PFs from hHTX were lower than those in RH4, RH7, and RH10 (*p* < 0.05, Figure 7B). However, the levels of active PAI-1 were generally comparable, suggesting that increased cleavage or reversion to latent PAI-1 could have occurred in the rabbits. This variation could reflect predictable differences between RH in the rabbit versus the human PFs. The inclusion of hemothoraces not associated with RH in the hHTX group may have also contributed to the differences, as only 7 of 20 subjects with hHTX had attributes strongly indicative of RH. While differences attributable to clinical variables of processing could also account for these findings, it is clear that elevated levels of PAI-1 and its activity were detectable in both the RH model and clinical hHTX PFs. The levels of active PAI-1 (Figure 7A), IL-8 (Figure 7C), and TNF-α (Figure 7D) observed in RH7 were comparable (*p* > 0.05) to those determined in the PFs of patients with hemothoraces. Concentrations of IL-6 (Figure 7E), glucose (Figure 7G), LDH (Figure 7H), and total protein (Figure 7I) were lower (*p* < 0.05) than those observed in humans with hemothoraces of different etiologies, likely reflecting the same influences or interspecies variability. The levels of proinflammatory biomarkers and PAI-1 observed in the RH7 model were higher than those for chemically induced pleural injury in rabbits and lower than the same biomarkers in the previously reported rabbit model of acute empyema [31]. Notably, the presence of elevated PAI-1 activity serves to suppress intrapleural fibrinolysis, suggesting that this finding likely promotes intrapleural organization associated with RH in all the rabbit models. Overall, the presence of active PAI-1 offers a compelling rationale to supplement the fibrinolytic activity in the pleural compartment, as its presence in RH PFs impairs local fibrinolysis.

There is currently no consensus about whether IPFT, otherwise called intrapleural lytic therapy, should be initially deployed in RH, what agents should be preferred, or how it should be optimized for use in non-surgical candidates for treatment [10,39]. Similar to the challenges in empyema management, the question of which intrapleural fibrinolysin is optimal for the treatment of RH and what dose and schedule of plasminogen activator should be administered, remains unknown [10,20]. Since the rabbit fibrinolytic system closely approximates that of humans, human fibrinolysins such as single (sc) or two (tc) chain uPA or tPA can be tested in the RH7 model.

The rabbit models of TCN-induced pleural organization and empyema have informed clinical trial testing and regulatory approval of scuPA in empyema [17]. We likewise infer that the RH model can similarly be used to define optimal doses and administration schedules that can inform clinical trial testing and thereby enable more efficient, evidence-based regulatory approval by agencies such as the FDA. The RH model also is amenable to interventional testing at different stages of disease. Our findings demonstrate that interventional effects can be tracked over time using state-of-the-art ultrasonography and CT imaging in addition to more traditional gross inspection of tissue and PF analyses. Other reported models may be less tractable, more expensive, of unclear reliability or safety, and they may be less suitable for interventional testing in relatively large numbers or stratification by stage of intervention, age, or sex.

## 4. Materials and Methods

### 4.1. Animals

All protocols were approved by the University of Texas at Tyler Institutional Animal Care and Use Committee (IACUC) before the study was initiated. New Zealand White pathogen-free rabbits were purchased from Charles River (Wilmington, MA, USA). Conventional animal husbandry practices were used to house rabbits in the UTHSCT vivarium, including plastic slotted-bottom cages, conventional rabbit feed, and water by bottle. Environmental and food enrichment were provided during the duration of the experiment. All personnel involved in animal husbandry, handling, procedures, anesthesia, or euthanasia underwent training endorsed by the veterinarian and IACUC. All agents administered to the rabbits were of pharmaceutical grade and thrombin (Sigma-Aldrich, Saint Louis, MO, USA) was culture tested. Rabbits were housed and acclimated for one week prior to the experiment. The rabbits were allocated for non-survival (terminal) blood collection (described below) or for induction of hemothorax model based on their capacity to tolerate the vest (Table 1).

### 4.2. Non-Survival Blood Collection

Donor rabbits were anesthetized in accordance with the IACUC-approved SOP manual for Perioperative Care of Rabbits. The left thorax was surgically prepared as described in the Vivarium SOP. An 18-gauge, 1-inch needle attached to an extension set was inserted between the left 5th–7th intercostal space (at point of palpable heartbeat) and 10–30 mL citrated syringes (1:7 citrate to blood ratio) were used to slowly aspirate blood from the heart. If needed, to expedite blood collection, an immediate thoracotomy was alternatively performed to collect additional blood directly from the exposed heart. Following exsanguination, at least 1 mL of an approved, commercial, euthanasia solution (sodium pentobarbital 390 mg/mL and phenytoin 50 mg/mL, Akorn Animal Health, Lake Forest, IL, USA) was delivered directly into the heart. Each rabbit was then monitored for cessation of heartbeat, spontaneous breathing, and corneal reflexes to confirm death.

### 4.3. Anesthesia Protocols

All sedation/anesthesia was administered according to UTHSCT vivarium standard operating procedure manual (SOP) and under the direction of the attending veterinarian and/or approved vivarium personnel. Buprenorphine (Stephenson’s Pharmacy, Tyler, TX, USA) 0.05 mg/kg IM (0.03–0.1 mg/kg) and Midazolam (Avet Pharmaceuticals Inc., East Brunswick, NJ, USA) 0.5 mg/kg IM (0.05–2.0 mg/kg) were used for pre-anesthesia sedation and pain medication. Induction of anesthesia was accomplished using dexmedetomidine (Zoetis Animal Health, Parsippany-Troy Hills, NJ, USA) 0.05 mg/kg IM (0.02–0.1 mg/kg), and Ketamine (Zoetis Animal Health, Parsippany-Troy Hills, NJ, USA) 50 mg/kg (20–60 mg/kg) was added for donor rabbits, which did not receive isoflurane. Local anesthesia was achieved using Lidocaine (Covetrus, Columbus, OH, USA) 1 mg/kg and Bupivacaine (Hospira, Lake Forest IL, USA) 1 mg/kg locally as dermal and intercostal block. Isoflurane (Covetrus, Columbus, OH, USA) 1–2% (0.5–5%) was used for maintenance of anesthesia. A reversal agent, Atipamezole (Zoetis Animal Health, Parsippany-Troy Hills, NJ, USA) up to 10 mg per 1 mg of dexmedetomidine (0.02–1 mg/kg) SQ or IM, was used as needed. Dosing schedules used in this study were vetted from over thirty years in which we have developed pleural injury models in the rabbit [14], particularly over the past fourteen years with current veterinary support. The dosing used was designed to mitigate adverse events and ensure effective sedation. Ketamine was used only in non-surviving donor rabbits. There were no apparent adverse drug interactions. The veterinarian was available for all experiments and for dose adjustments as required. The effects of all agents in each individual rabbit were evaluated by the attending veterinarian and/or approved vivarium personnel.

### 4.4. Post Procedure Analgesics, Support, and Monitoring

Rabbits were monitored continuously during surgical procedures and at least every 12 h for the first 2 days post procedure and then daily until the end of experiment. Physiologic assessments included respiratory rate and effort, heart rate, oxygen saturation assessed by pulse oximeter, fecal and urine volume, body temperature, and behavioral indices including agitation or lassitude. Post-surgical support included external heat, positioning to improve respirations, oxygen supplementation, subcutaneous fluids, gel cups, and pain medications. Pain tolerance and need for analgesics varied in individual rabbits. Dosage and frequency of administration were individualized to ensure comfort. Environmental and food enrichments were used to distract rabbits from removing protective jackets or dislodging chest tubes.

### 4.5. Humane Endpoints

The following criteria were used in determining humane endpoints requiring euthanasia. These included inability to maintain sPO2 of greater than 85% on room air, a heart rate greater than 360 beats per minute (bpm), extreme lethargy or weakness indicated by an inability to maintain sternal position or inability to support the head in a customary normal position above the chest, respiratory distress as indicated by increased respiratory rate (greater than 60 breaths per minute) or dyspnea as indicated by increased effort and open mouth breathing, and inability to maintain body temperature above 99 °F. Once rabbits demonstrated any of the above signs, vivarium staff and/or the veterinarian was notified. Additional support measures including oxygen supplementation, external heat, and additional pain medication were used as needed and at the recommendation of the veterinarian. If there was no salutary response to treatment, the rabbit was euthanized by intravenous or intracardiac injection of a commercial euthanasia solution as per the SOP Rabbit Euthanasia protocol by vivarium staff, attending veterinarian, or approved personnel.

### 4.6. Chest (Thoracostomy) Tube Placement Procedure

The thoracostomy tube was a 12-gauge, 20 cm, single lumen, radiopaque, polyurethane catheter with multiple fenestrations 6 cm from distal tip (Mila International, Florence, KY, USA) including a J-tip guidewire system distributed for veterinary medical use. The thoracostomy tube was placed under general anesthesia plus local anesthesia after surgical preparation of the right thorax from the 4th to the 12th rib. Using sterile technique, a 2–3 cm vertical skin incision was made, dorsal to the dorsal plane and between rib spaces 5–8. Using blunt dissection, the deep tissue over the desired rib space was exposed. A sterile, metal feeding needle (16 gauge) was inserted through the deep intercostal muscle and pleura. A J-tip guidewire was then inserted within the lumen of the feeding needle and the guidewire was advanced into the pleural cavity to desired depth. The feeding needle was then carefully removed from the thorax, leaving the guidewire in the pleural space. The tip of the chest tube was placed over the wire and the chest tube advanced through the skin and muscle into the pleural space with gentle pushing. The chest tube was positioned over the guidewire until all the fenestrations were within the pleural space. The guidewire was then removed from the chest tube, the tube clamped, and a valve attached to distal end of chest tube. The clamp was released with evacuation of any air from the pleural space by aspiration with a syringe until negative pressure was established. A purse string suture was then placed around the insertion of the chest tube in the deep musculature using 4-0, non-braided, absorbable suture material (PDS or PDS plus). The skin incision was closed using 3-0, non-braided, absorbable suture material (PDS or PDS plus) in an interrupted suture pattern. A horizontal mattress suture was placed incorporating the chest tube, after which the tube was secured with a modified Roman sandal pattern. The tube was further secured to the skin with sutures. The chest cavity was aspirated a second time to eliminate any remaining air in the pleural space and restore the negative pressure in the pleural space. The insertion site was covered with a sterile gauze dressing and other protective wound covers including a cloth jacket and non-adhesive dressing. The bandaging was loose enough to avoid impediment of the animal’s ability to breathe. The tube was then flushed with sterile saline at least once daily and after infusion of blood. Thoracic ultrasonography or chest CT imaging was performed to confirm correct placement of the tube. If the chest tube was dislodged, it was removed by cutting remaining sutures with retraction from the chest followed by manual pressure. A clean dressing was then applied. If pneumothorax was detected, it was aspirated via the chest tube or alternatively through a 22-gauge needle attached to syringe and/or extension sets under the supervision of vivarium staff or veterinarian.

### 4.7. Induction of RH

Citrated homologous whole rabbit blood (30–60 mL) supplemented with 0.3 U/mL thrombin (Sigma-Aldrich, Missouri, MO, USA) and 10 mM CaCl_2_ (American Regent, Inc., Shirley, NY, USA) was infused into the pleural space through the chest tube a total of 3 times over 72–80 h to simulate the clinical presentation of continuous bleeding. Recalcification and thrombin supplementation of administered blood were performed to initiate intrapleural clot formation immediately after administration, as determined in ex vivo clotting experiments using a fibrometer. Recalcification and thrombin supplementation mitigated clot resorption and promoted relatively large RH collections that were durable and apparent by gross examination for up to 10 days after induction. The initial dose was 30 mL of blood, followed by an additional dose of 60 mL of intrapleural blood at 24 h and 30 mL at 96 h, as tolerated. After each procedure, rabbits were evaluated for respiratory distress with imaging and monitored for recovery. The protocol is illustrated in Figure 1.

### 4.8. Ultrasonographic Imaging and 2D Clot Morphometry

Progression of RH was monitored daily via B-mode ultrasonography, utilizing R5.2.x software, multifrequency transducer model 12L-RS, at a frequency of 10 MHz. Ultrasonography was tolerated by awake animals. Using the same software, a 2D sonograph of a clot was developed and standardized over time, serving as a reference for the 2D clot area (cm^2^) measurements at RH4, RH7, and RH10 endpoints.

### 4.9. Computed Tomography and Lung Restriction Evaluation

Computed tomography was performed on baseline, post-chest tube placement, post RH induction, and before euthanasia. Raw lung volume calculations and CT image processing of the RH4, RH7, and RH10 models were performed using Vivoquant Studio 3.0.

### 4.10. Euthanasia and Postmortem Evaluation

All rabbits within the RH4, RH7, and RH10 were euthanized with at least 1.0 mL commercial euthanasia solution given IV. Gross presentation of the pleural cavity and retained clot were evaluated and photographed. To avoid bias, pleural adhesion was evaluated by a surgeon (A.O.A) not involved in the development of the RH models. The rabbit lungs were harvested and perfused with 20 mL of 10% formaldehyde for 72 h fixation.

### 4.11. Histology

The excised and fixed lung tissues were paraffin-embedded, sectioned into 5 µm, and attached into adhesive slides. H&E (StatLab, McKinney, TX, USA) staining visualized the pleural thickening (1 mm scale) while Masson Trichrome (ThermoFisher, Waltham, MA, USA) technique demonstrated the overexpression of collagen (1 mm scale) in the pleural lining at 4× magnification. Color Brightfield imaging mode and morphometric analysis of pleural thickening were performed with Cytation 10 (Biotek, Winooski, VT, USA) and Gen 5 v3.11 software, respectively.

### 4.12. Inflammatory Profile and Biomarker Quantification

ELISAs was used to measure the levels of the following inflammatory markers in RH4, RH7, and RH10 and hHTX PFs: Active and total PAI-1 (Innovative Research, Pleasanton, CA, USA), IL-6 (R&D Systems, Minneapolis, MN, USA), IL-8 (RayBiotech, Peachtree Corners, GA, USA), TNF-α (R&D Systems, Minneapolis, MN, USA), and, TGF-β (R&D Systems, Minneapolis, MN, USA). Glucose and protein concentration and LDH activity in PFs were evaluated by glucose colorimetric/fluorometric kit (Sigma-Aldrich, Missouri, MO, USA), LDH Activity kit (Sigma-Aldrich, Missouri, MO, USA), and Pierce™ BCA assay kit (ThermoFisher, Waltham, MA, USA), respectively.

### 4.13. Cytological Evaluation: RBC and WBC Counts with Differentials

Total RBC and WBC counts and differential analyses were performed as previously reported [31].

### 4.14. Human RH Fluids

These samples were obtained from the trauma surgery group at UT Health East Texas under an approved IRB protocol (2021-003). The demographics of the clinical cohort of patients including those with RH attributes are described in Table 2.

### 4.15. Statistics

Based on previous publications, this study was sufficiently powered to reliably evaluate the outcomes using six rabbits per group. Kruskal–Wallis (Nonparametric, One-Way ANOVA) was used to determine the statistical significance and to compare the mean ranks of each dataset with more than two groups, adjusting the p-value with Dunn’s multiple comparison test. By assuming α level of significance of 5%, there is greater than 80% power to detect the mean 2D clot area and pleural adhesions difference of 0.48 SD with six subjects per group. Mann–Whitney test (Nonparametric, t-test) determined statistical significance of dataset with only two groups, following similar assumptions as mentioned earlier. Data were presented as box plots while the graphs were generated using GraphPad Prism v 9.3.1.

## 5. Conclusions

We report a novel model of RH in the rabbit in which durable chest tube placement can be achieved and in which pleural organization is characteristic. The model can support testing of the efficacy and safety of interventional approaches and in particular IPFT, with or without adjuncts like DNase [40] or PAI-1-TFT with antibodies [30] or peptides. We plan to use the model in the future to optimize pharmacologic interventions in anticipation of clinical trial testing. We hope that broad use of this model may encourage other groups to likewise test novel interventional strategies to improve pleural organization and outcomes of RH.

## Figures and Tables

**Figure 1 ijms-25-00470-f001:**
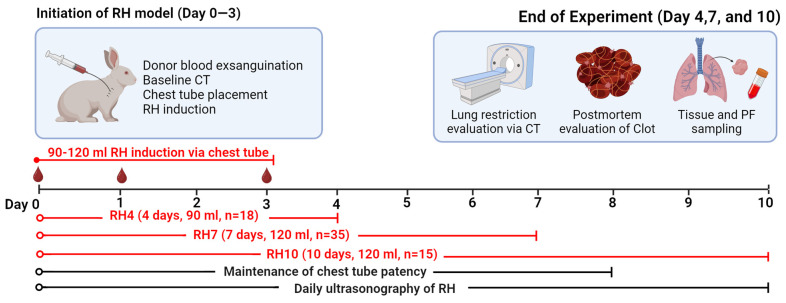
Schematic representation of the induction and assessments of the RH model. The temporal development of the model and the number of rabbits used in the 4 (RH4), 7 (RH7), and 10 (RH10) days of the study are displayed. Chest tube placement was performed on Day 0 at the initiation of the RH model (Day 0–3), with the red droplets representing each delivery of blood via the chest tube. Techniques used to evaluate lung restriction and clot retention on the final day of the model development are shown on the top right panel. Postmortem pleural fluid and lung tissues were evaluated by histologic and biochemical analysis. This figure was created using Biorender.com.

**Figure 2 ijms-25-00470-f002:**
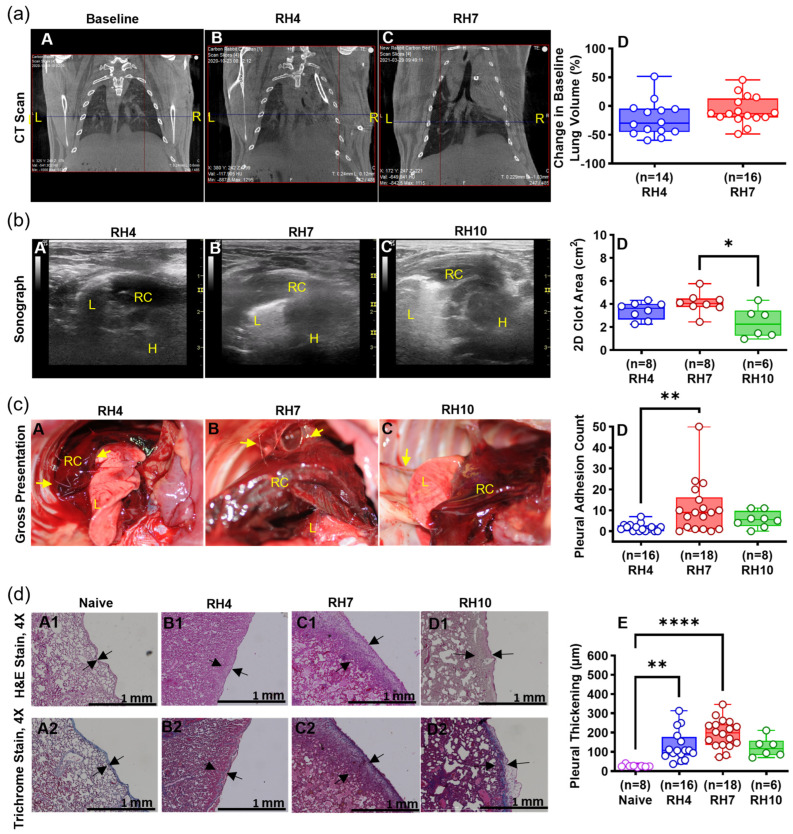
Progression of pleural injury in the RH model. (**a**) CT images showing baseline non-injured lungs (**A**) and extensive opacification of the right; ipsilateral hemothorax in which RH was induced; moreover, opacification persists in the RH4, RH7, and RH10 models with limited resolution related to a combination of pleural fluid, atelectasis, or RH. (**B**,**C**). A negative change in % indicates a decrease in baseline lung volume at each model endpoint (**D**). No CT data were available for RH10 due to technological issues precluding imaging. L = Left; R = Right. (**b**) Representative ultrasonographic images display the retained clot (RC), lung (L), and heart (H) at each endpoint (**A**–**C**). Morphometric analysis of sonographs from each model was compared (**D**). (**c**) Gross presentation of the pleural cavity was examined, showing the retained clot (RC), lung (L), and pleural adhesions (**A**–**C**, yellow arrows). The pleural adhesion counts included fibrin strands, webs, and sheets which were indicative of pleural organization (**D**). (**d**) Naïve and injured rabbit lung tissue were stained with Hematoxylin and Eosin (**A1**–**D1**, H&E) and Masson Trichome Stain (**A2**–**D2**, Trichrome) to demonstrate pleural thickening and collagen deposition, respectively. Black arrows indicate the thickening in the pleural lining. Blue-colored stain represents the collagen deposition in the pleural lining (**A2**–**D2**). All histological images were imaged at 4× objective with a 1 mm scale bar. Morphometric analysis compared the pleural thickening between naïve and RH injured groups (**E**). Mann–Whitney (panel (**d**), **E**) and Kruskal–Wallis with Dunn’s multiple comparison (panel (**b**), **D**; panel (**d**), **E**; panel (**d**)) tests determined the statistical significance of the datasets, with *, **, and **** representing *p* < 0.05, *p* < 0.01, and *p* < 0.0001, respectively.

**Figure 3 ijms-25-00470-f003:**
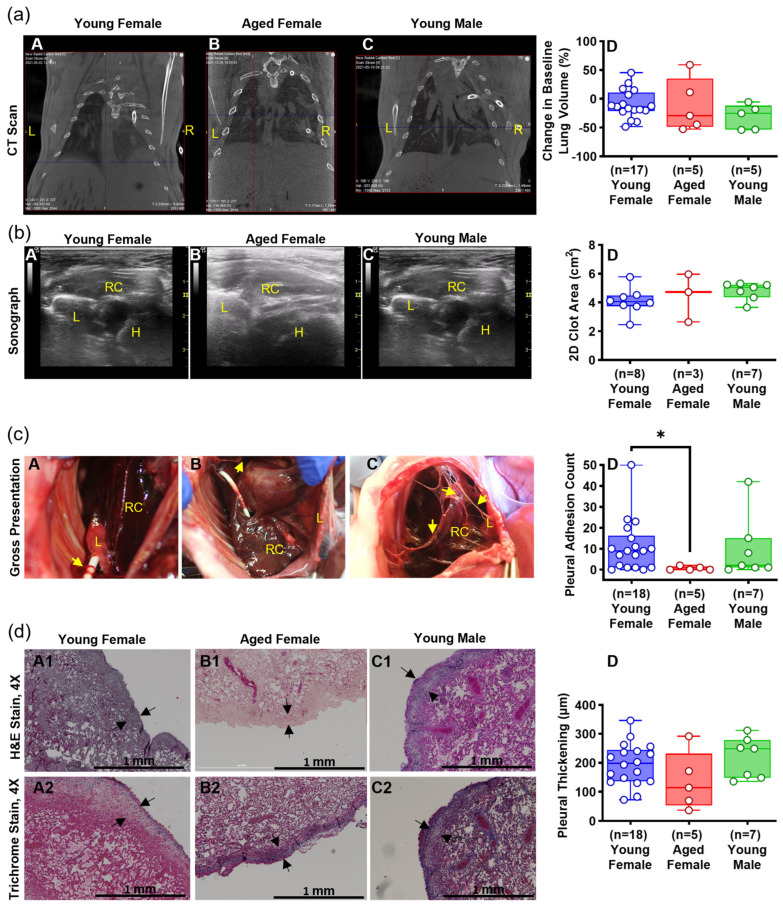
Effects of sex and age on the development of RH at 7 days. Similar to our initial RH7 experiments with young female rabbits (*n* = 22), RH was also induced in young male (*n* = 8) and aged female (*n* = 5) rabbits to determine the contribution of age and sex to RH development. (**a**) CT images comparing the opacification at right hemithorax between age/sex (**A**–**C**). Percent changes in the baseline lung volume of rabbits with RH for 7 d were evaluated between age/sex (**D**). L = Left and R = Right. (**b**) Ultrasonography at 7 d was performed on all age/ender groups to detect the lung (L), retained clot (RC), and heart (H). 2D Clot area at the 7 d endpoint was measured and compared between the age/sex groups (**A**–**D**). (**c**) Gross examination of the lung (L) retained clot (RC) and pleural adhesions (yellow arrows) was evaluated to confirm the ultrasonographic readings (**A**–**C**). As previously mentioned, the presence of pleural adhesions such as fibrin strands, sheets, and webs was indicative of pleural injury (**D**). (**d**) Histology of the injured lung tissue was compared between the age and sex groups. Hematoxylin and Eosin (H&E, **A1**–**C1**) staining and imaging at 4× demonstrate pleural thickening (black arrows, 1 mm scale). Morphometry demonstrates the differences in pleural thickening between age/sex groups (**D**). Staining with Masson’s Trichrome (**A2**–**C2**) suggests overexpression of collagen (blue fibrillar material) associated with peripheral atelectasis of the lung parenchyma. Representative images are illustrated in each panel, as the findings were observed in all rabbits in these groups. Kruskal–Wallis test with Dunn’s multiple comparison test demonstrated statistical difference with * representing *p* < 0.05.

**Figure 4 ijms-25-00470-f004:**
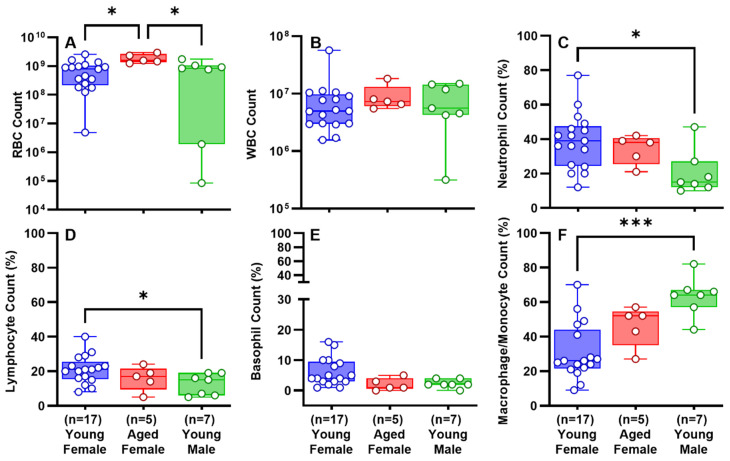
Cellular analysis and evaluation of the PFs of young and aged rabbits with RH for 7 days. Total red blood cell (RBC) (**A**) and white blood cell (WBC) (**B**) counts with differentials (**C**–**F**) were collected and analyzed from young and aged rabbit PFs with RH for 7 days. Kruskal–Wallis test with Dunn’s multiple comparison test was used to determine statistical significance of the dataset, representing *p* < 0.05 as * and *p* < 0.001 as ***.

**Figure 5 ijms-25-00470-f005:**
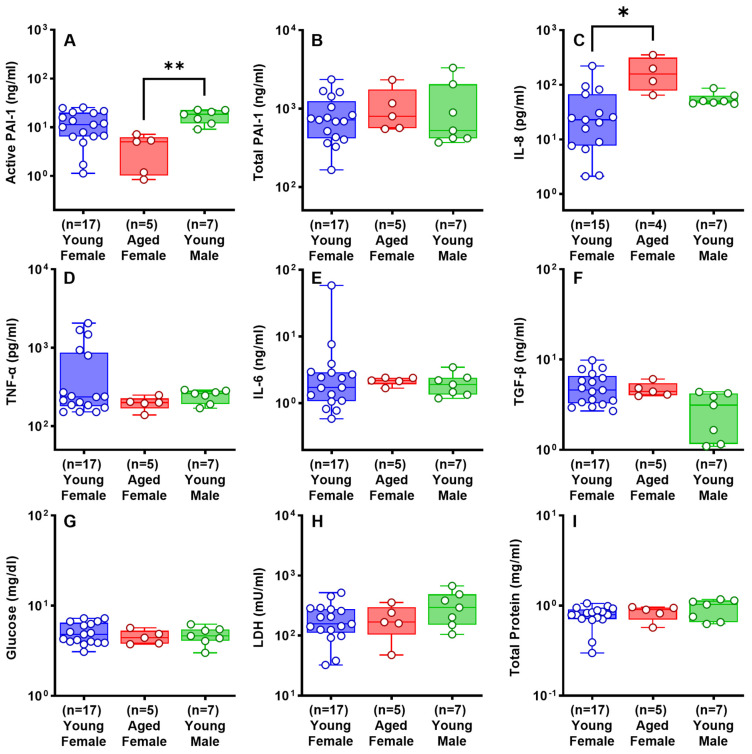
Analysis and comparison of the biomarkers of inflammation in the PFs of young and aged rabbits with RH for 7 days. The following markers of inflammation were quantified: Total (**A**) and active (**B**) PAI-1, TNF-α (**C**), IL-8 (**D**), IL-6 (**E**), TGF-β (**F**), Glucose (**G**), LDH activity (**H**), and total protein (**I**). Kruskal–Wallis with Dunn’s multiple comparison test determined the statistical significance of the datasets, with *, **, representing *p* < 0.05 and *p* < 0.01, respectively.

**Figure 6 ijms-25-00470-f006:**
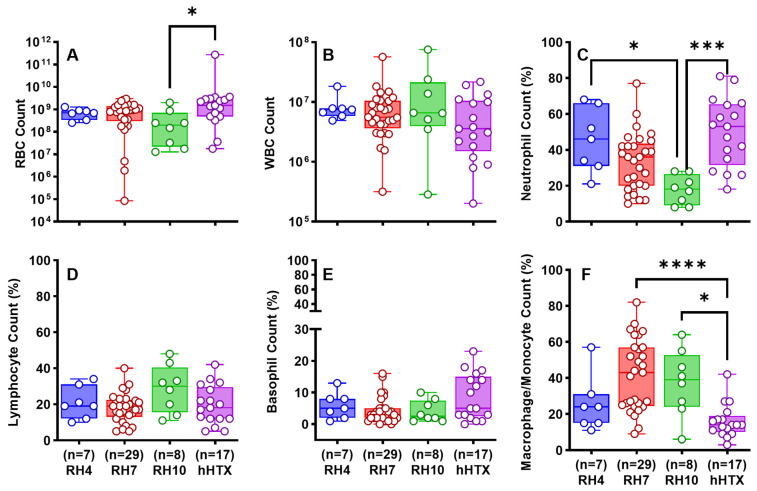
Comparison of the cellular profiles of RH rabbit models and traumatic hHTX PFs. Red (RBCs, (**A**)) and White Blood (WBCs, (**B**)) cell counts and WBC differentials (**C**–**F**) of rabbit RH models and human hemothorax (hHTX) PFs were evaluated as previously described [30,31]. The lack of PF samples and technological difficulties impeded data collection in RH4, RH7, and RH10 models and clinical subjects. Kruskal–Wallis with Dunn’s multiple comparison tests determined the statistical significance of the datasets, with *, ***, and **** representing *p* < 0.05, *p* < 0.001, and *p* < 0.0001, respectively.

**Figure 7 ijms-25-00470-f007:**
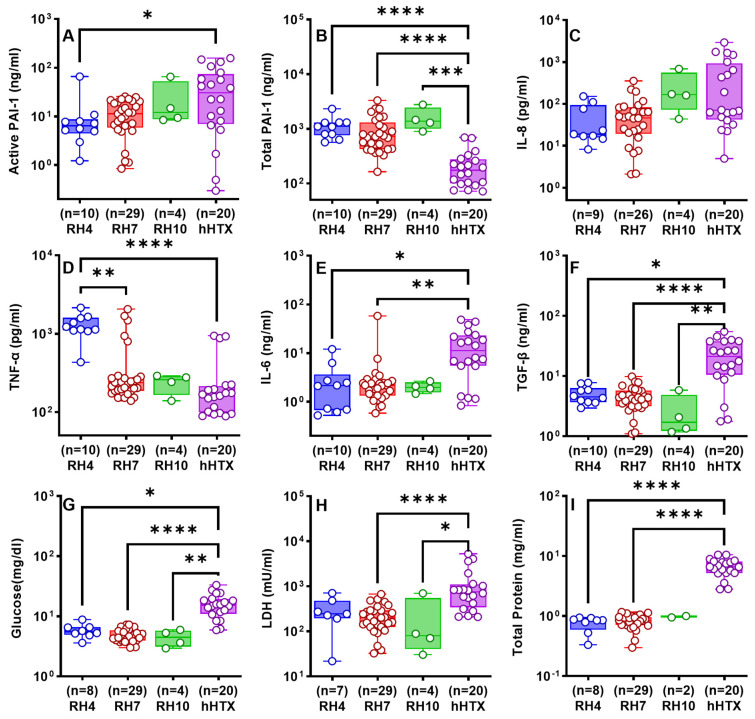
Comparison between the inflammatory and biochemical profile of RH rabbit models and hHTX PFs. Total (**A**) and active (**B**) PAI-1, TNF-α (**C**), IL-8 (**D**), IL-6 (**E**), TGF-β (**F**), Glucose (**G**), LDH activity (**H**), and total protein (**I**) in PFs of rabbit RH models and traumatic hHTX were assayed as previously described [14,30]. Likewise, data point collection in the rabbit RH models and clinical subjects was impeded by the lack or minimal amount of pleural effusion in the model (<1 mL). Prominently, RH4 and RH10 PF samples were unavailable, elucidating the lack of glucose (RH4, *n* = 2, (**G**)), LDH activity (RH4, *n* = 3, (**H**)), and total protein (RH4, *n* = 2; RH10, *n* = 10, (**I**)) data points. IL-8 data points (RH4, *n* = 1; RH7, *n* = 3) that were below the detection level (<1.9 pg/mL) were excluded from analysis (**C**). Kruskal–Wallis with Dunn’s multiple comparison tests determined the statistical significance of the datasets, with *, **, ***, and **** representing *p* < 0.05, *p* < 0.01, *p* < 0.001, and *p* < 0.0001, respectively.

**Table 1 ijms-25-00470-t001:** Outcome metrics of the RH model groups, assessments, and clinical presentation. The data below represent all the animal subjects used during RH model development. CT scans were not obtained in the RH10 model due to technological issues.

Retained Hemothorax Model Groups, Outcomes			
Experimental Group	RH4	RH7	RH10
Young female	18	22	8
Young male	0	8	7
Aged female	0	5	0
Donor Group			
Young female	16	42	17
Successful induction of RH	16/18 (88.9%)	30/35 (85.7%)	15/15 (100%)
Chest tube placement success	15/18 (83.3%)	32/35 (91.4%)	15/15 (100%)
Pneumothorax incidence at endpoint	0/16 (0%)	1/30 (3.3%)	No CT
Pneumothorax volume aspirated (mean ± SD, mL)	58.5 ± 23.2	70.0 ± 16.1	70.0 ± 20.3
RH model Outcomes			
Change in lung volume at endpoint (mean ± SD, %)	−22.29 ± 30.59	−3.29 ± 24.10	No CT
Estimated 2D clot area at endpoint (mean ± SD, cm^2^)	3.43 ± 0.75	4.43 ± 0.96	2.43 ± 0.96
Pleural adhesions (mean ± SD, count)	1.75 ± 1.88	9.07 ± 12.34	6.67 ± 4.47
Pleural thickening (mean ± SD, µm)	130.3 ± 78.40	191.1 ± 78.53	156.0 ± 121.5
Estimated pleural effusion volume (mean ± SD, mL)	4.92 ± 4.17	5.34 ± 4.34	0.78 ± 0.73
Rabbits with pleural effusion (%)	7/16 (43.8%)	29/30 (96.7%)	7/15 (46.7%)

**Table 2 ijms-25-00470-t002:** Clinical characteristics of UTHSCT patients with traumatic hemothorax. PFs were collected from patients with traumatic hemothorax either during day of admission or variably at different days hospitalization as clinically indicated.

Demographics of UT Health East Texas Patients with Traumatic Hemothorax	
Sample Size	*n* = 20
Age, mean ± SD	50.2 ± 21.4
Age, categorical	
<45	7
45–64	9
65–80	3
>80	1
Race, categorical	
Non-Hispanic White	15
Non-Hispanic Black	2
White/Hispanic	1
Unknown	2
Sex	
Male	14
Female	6
Mechanism of Injury	
Blunt Trauma (MVC, GLF or FFH, Auto-Ped)	17
Penetrating Trauma (GSW to Chest)	3
Chest-Related Complications	
Single or Multiple Rib FXs	17/20 (85%)
Atelectasis	10/20 (50%)
Empyema	None
Pneumothoraces	16/20 (80%)
Outcomes	
Days of Hospitalization (mean ± SD)	17.3 ± 14.8
Mortality within 14 days/up to a year post discharge	6/20 (30%)
Hemothorax Reoccurrence	1/20 (5%)
Resolution w/chest tube or pig-tail catheter drainage	13/20 (65%)
Failed resolution with drainage (referral to surgery)	7/20 (35%)

## Data Availability

All data were shared according to the NIH Resource Sharing statements per funded grants.

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
