# Peer review of "A Novel Rabbit Model of Retained Hemothorax with Pleural Organization"

_ijms, 2023, doi:10.3390/ijms25010470_

Round 1

Reviewer 1 Report

Comments and Suggestions for Authors

The manuscript entitled "A Novel Rabbit Model of Retained Hemothorax with Pleural Organization" deals with the development and the validation of a new model of retained hemothorax in rabbits. This manuscript is well written, the introduction is concise and clearly explains the reasons behind the investigation of the topic and the potential interests in this research field. Citations are well placed. The conclusions are well supported by the experimental part and tables and figures are very descriptive. Self-citations have been corrected in the revised pdf (attached file)

Nothing to add, I recommend to accept the manuscript in the present form

Author Response

Comments: The manuscript entitled "A Novel Rabbit Model of Retained Hemothorax with Pleural Organization" deals with the development and the validation of a new model of retained hemothorax in rabbits. This manuscript is well written, the introduction is concise and clearly explains the reasons behind the investigation of the topic and the potential interests in this research field. Citations are well placed. The conclusions are well supported by the experimental part and tables and figures are very descriptive. Self-citations have been corrected in the revised pdf (attached file)

Nothing to add, I recommend to accept the manuscript in the present form.

Response: We very much appreciate the favorable review, impression that the work was clearly presented, results well-supported and recommendation to accept as originally submitted.

Reviewer 2 Report

Comments and Suggestions for Authors

The topic of the study is interesting because the authors propose a new model of retained hemothorax in rabbits.

The manuscript is overall well written. The structure results properly organized.

 I suggest the authors to make the abstract more systematic. I think it is too detailed.

The introduction is overall good.

The methodological approach is correct. Images, table and descriptions are clear.

My point of view it is mandatory to have section no. 5 Conclusions. This article does not contain 5. Conclusions, which makes it difficult for readers.

Author Response

Comments: The topic of the study is interesting because the authors propose a new model of retained hemothorax in rabbits.

The manuscript is overall well written. The structure results properly organized.

 I suggest the authors to make the abstract more systematic. I think it is too detailed.

The introduction is overall good.

The methodological approach is correct. Images, table and descriptions are clear.

My point of view it is mandatory to have section no. 5 Conclusions. This article does not contain 5. Conclusions, which makes it difficult for readers.

Responses: 1) We appreciated the determination that the study was of interest and that 2 and 3) the manuscript was well-written and properly organized.

4) Abstract: We offer that a large amount of data was described in this manuscript and respectfully submit that the Abstract describes key elements which enable the reader to understand the work in summary form, with specifics in our response to the Editor to point 2.

5 and 6) We appreciate the determination that the Introduction was good, the methodology appropriate and the data presentation was clear.

7) We thank the reviewer for this suggestion and have added a new Section 5; Conclusions which now follows the Discussion to reiterate the key findings.

Reviewer 3 Report

Comments and Suggestions for Authors

The article does not fit the scope of the International Journal of Molecular Sciences, where molecules are the object of study.

The authors are describing a new animal model of retained hemothorax. However they did not perform any "omics" studies (genome, RNA, protein, metabolites), nor did they conduct any extensive experiments that had molecules as the main tool to approach one of the three main categories of studies adequate for this journal:

  • Fundamental theoretical problems of broad interest in biology, chemistry and medicine;
  • Breakthrough experimental technical progress of broad interest in biology, chemistry and medicine;
  • Application of the theories and novel technologies to specific experimental studies and calculations.

They did quantify markers of inflammation: total and active PAI-1, TNF-α, IL-8, IL-6, TGF-β, glucose, LDH activity, and total protein. But this does not constitute an extensive investigation of inflammation, which is perhaps not their aim.

I suggest that the editors transfer the article to a MDPI journal more adequate to the scope of this manuscript.

Author Response

Comments: The authors are describing a new animal model of retained hemothorax. However they did not perform any "omics" studies (genome, RNA, protein, metabolites), nor did they conduct any extensive experiments that had molecules as the main tool to approach one of the three main categories of studies adequate for this journal:

  • Fundamental theoretical problems of broad interest in biology, chemistry and medicine;
  • Breakthrough experimental technical progress of broad interest in biology, chemistry and medicine;
  • Application of the theories and novel technologies to specific experimental studies and calculations.

They did quantify markers of inflammation: total and active PAI-1, TNF-α, IL-8, IL-6, TGF-β, glucose, LDH activity, and total protein. But this does not constitute an extensive investigation of inflammation, which is perhaps not their aim.

I suggest that the editors transfer the article to a MDPI journal more adequate to the scope of this manuscript.

Response: We respectfully counter that we think this work meets criteria 1 for this journal, as the creation of a model of retained hemothorax was deemed by the National Heart Lung and Blood Institute of the National Institutes of Health to be important enough to be funded and met an important gap in the field as summarized in the Abstract and described further in the Introduction. We cited NIH funding of this study and peer reviewers of our NIH grant application endorsed this project after competitive review. We also feel that this work meets criteria 2 for the scope of this journal as we describe a new technique to establish this model in the rabbit and thereby advance the field, as described in the Abstract, Introduction Methods and Discussion. Lastly, we respectfully wish to point out that the other reviewers and editorial decision vary from this perspective about the suitability of this manuscript as to consideration for publication in this journal.

Reviewer 4 Report

Comments and Suggestions for Authors

Thanks for the invitation to review this important study. I like reading how they tried the development and validation rabbit model. The authors have discussed that they follow anesthesia protocols and administered Buprenorphine, Midazolam, Dexmedetomidine, and Ketamine. While these drugs are commonly used in rabbit anesthesia, could you provide additional information on how the chosen dosages were determined, considering the potential variations in individual rabbit responses and the need for balancing effective sedation with minimizing adverse effects? Additionally, was there any consideration given to potential drug interactions within this combination? 

Authors mention that they infuse citrated homologous whole rabbit blood into the pleural space. Could you elaborate on the rationale behind supplementing the blood with thrombin and CaCl2, and how this infusion protocol simulates continuous bleeding in the clinical context?

Author Response

Comments: Thanks for the invitation to review this important study. I like reading how they tried the development and validation rabbit model. The authors have discussed that they follow anesthesia protocols and administered Buprenorphine, Midazolam, Dexmedetomidine, and Ketamine. While these drugs are commonly used in rabbit anesthesia, could you provide additional information on how the chosen dosages were determined, considering the potential variations in individual rabbit responses and the need for balancing effective sedation with minimizing adverse effects? Additionally, was there any consideration given to potential drug interactions within this combination? 

Authors mention that they infuse citrated homologous whole rabbit blood into the pleural space. Could you elaborate on the rationale behind supplementing the blood with thrombin and CaCl2, and how this infusion protocol simulates continuous bleeding in the clinical context?

Responses: We appreciate the favorable comments. We also appreciate the excellent questions about the dosing choices. The dosages of the agents used have been refined over the course of the more than thirty years during which we have developed and used rabbit models of tetracycline induced pleural injury and empyema involving different agents and durations of infection (Ref 14 of the manuscript). Ketamine was only administered in donor (non-surviving) rabbits, which did not receive Isofluorane (Line 453). The veterinarian developed the protocols that were applied in this manuscript over the past 14 years, was always available for the procedures and was responsible for making dose adjustments as needed. This unique experience enabled us to use the dosing and administration schedules we described in the Methods.  We have not detected drug interactions alter administration of the drugs as we described. We added additional text on Lines 457-463 of renumbered Section 5.3 of the Methods to address these points.

As our objective was to generate RH in the pleural space, we added thrombin and recalcified the blood to promote intrapleural clotting. Both measures are established techniques to promote clotting of citrated blood, which is how the blood was collected from the donor rabbits (Methods subsection now 5.2., non-survival blood collection). In early pilot experiments, we found that administration of intrapleural blood without these adjuncts did not reliably allow formation of relatively large organizing thrombus in the pleural space. To encourage formation of the intrapleural thombus, we sought to strongly initiate coagulation right after administration of blood through the thoracostomy tube, which is why the more rapidly acting thrombin was added. We tested the combination of recalcification and thrombin supplementation to initiate clotting ex vivo in a device called a fibrometer in preliminary experiments, found that coagulation was initiated immediately after intrapleural delivery of the supplemented blood and used the strategy described in the Methods accordingly.  We added new text on Lines 532-537 in the renumbered section 5.7 of the Methods; Induction of RH, to address these excellent points.

Round 2

Reviewer 2 Report

Comments and Suggestions for Authors

Accept